

# First steps towards the reconstruction of the squark flavour structure

Jordan Bernigaud[*] and Björn Herrmann[†]

Univ. Grenoble Alpes, Univ. Savoie Mont Blanc, CNRS, LAPTh, F-74000 Annecy, France

[*] bernigaud@lapth.cnrs.fr, [†] herrmann@lapth.cnrs.fr

## Abstract

Assuming the observation of a squark at the Large Hadron Collider, we investigate methods to access its flavour content and thus gain information on the underlying flavour structure of the theory. Based on simple observables, we apply a likelihood inference method to determine the top-flavour content of the observed particle. In addition, we employ a multivariate analysis in order to classify different flavour hypotheses. Both methods are discussed within a simplified model and the more general Minimal Supersymmetric Standard Model including most general squark mixing. We conclude that the likelihood inference may provide an estimation of the top-flavour content if additional knowledge, especially on the gaugino sector is available, while the multivariate analysis identifies different flavour patterns and can accommodate a more minimalistic set of observables.


# 1 Introduction

One of the main goals of the Large Hadron Collider (LHC) is the quest for signals of physics beyond the Standard Model of particle physics. Its Run-2 being accomplished, however, no direct evidence pointing towards the existence of new states has been found. Among the numerous extensions addressing the shortcomings of the Standard Model, Supersymmetry ranks among the most attractive solutions. However, if Nature is indeed supersymmetric, it is reasonable to assume that its exact realization is situated beyond the "vanilla" Minimal Supersymmetric Standard Model (MSSM) or its simplified realizations which are typically searched for in current experimental analyses [1–15]. To give an example, experimental studies typically are based on the Minimal Flavour Violation (MFV) paradigm assuming that all flavour-violating interactions stem from the Yukawa couplings alone, as it is the case in the Standard Model. However, there is no apparent reason that this paradigm is respected beyond the Standard Model. In the MSSM, additional flavour-violating terms may be present in the Lagrangian, leading to a modified phenomenology. This possibility is labelled as Non-Minimal Flavour Violation (NMFV).

The assumption of NMFV in the squark sector has received considerable attention throughout the last decade. Numerous studies have addressed flavour precision observables [16–22], dark matter aspects [23–25], and most importantly collider signatures related to squark generation mixing [26–32]. In particular, it has recently been shown that non-minimal flavour mixing between the second and third generation squarks can easily be accommodated with respect to current experimental constraints from flavour and precision data [33–35]. Even more recently, it has become apparent that the current limits published by the ATLAS and CMS collaborations cannot directly be applied in such a configuration, but will be considerably weakened [36, 37]. In maximal mixing cases, squarks would even be likely to completely escape detection. Consequently, a dedicated search for characteristical signatures of non-minimal flavour violation in the squark sector is necessary. Such a strategy is proposed in Refs. [37] based on the search for mixed final states containing a top quark together with a charm-flavoured jet and missing transverse energy. In the following, we assume that this final state can be accessed with sufficient luminosity at the LHC as discussed in Ref. [37], allowing to include the currently uncovered parameter region.

Assuming the discovery of a squark-like state at the LHC, e.g., through the channel mentioned above, it will be crucial to understand its exact nature and in particular reveal its flavour content. This information will give important hints towards the flavour structure of the underlying theory and will hint towards possible Grand Unification frameworks [17, 38–48]. It is the main goal of the present Paper to investigate different methods for reconstructing the flavour content of an observed squark state. To simplify this first attempt, we concentrate on squarks containing top and charm flavour. This situation is less constrained by flavour and precision data [35] as compared to mixing with first generation flavours [17]. Moreover, squarks containing top flavour are easier to access from the experimental point of view. However, the methods presented in the present Paper are general and can be extended to the first generation or to the sectors of down-type squarks and sleptons.

Our study will rely on the pair production of a flavour-mixed squark [18] and its subsequent decays into either top or charm quarks plus missing transverse energy [29], or into bottom quarks and charginos. A direct reconstruction of the squark rotation matrix would basically be possible, provided that we have access to the corresponding branching ratios, potentially with the help of top-polarization measurements [49–52], plus complete information on the neutralino and chargino sector. In practice, having precise access to these information is not an option.

We therefore discuss methods aiming at inferring the top and charm content of the ob-

served squark and obtain information about the flavour structure requiring a minimal amount of prior knowledge. More precisely, we will apply two methods: the first based on a likelihood inference, the second relying on multi-variate analysis techniques. We emphasize that the present Paper does not aim at constructing a complete analysis, but rather show that these two methods may provide interesting approaches to the above question, provided complementary investigation.

The Paper is organized as follows: in Sec. 2, we review the model of our interest, namely the MSSM with NMFV in the squark sector. In Sec. 3 we discuss observables which are measurable at LHC and which we will base our analyses on. Sec. 4 is then devoted to the first method, a likelihood inference of the top-flavour content of the squark. The second method, the multivariate analysis, is then presented in Sec. 5 for the simplified setup before discussing it for the more realistic framework of the MSSM in Sec. 6. Finally, Sec. 7 contains our conclusions.

## 2 Model and parameters

As discussed in the Introduction, the model of our interest is the Minimal Supersymmetric Standard Model (MSSM) with $R$-parity conservation and the most general squark flavour structure. In the super-CKM basis, i.e. in the basis $(\tilde{u}_L, \tilde{c}_L, \tilde{t}_L, \tilde{u}_R, \tilde{c}_R, \tilde{t}_R)$, the hermitian up-type squark mass matrix can be written as [53]

$$\mathcal{M}_{\tilde{u}}^2 = \begin{pmatrix} V_{\text{CKM}} M_{\tilde{Q}}^2 V_{\text{CKM}}^\dagger + m_u^2 + D_{\tilde{u},L} & \frac{v_u}{\sqrt{2}} T_u^\dagger - m_u \frac{\mu}{\tan\beta} \\ \frac{v_u}{\sqrt{2}} T_u - m_u \frac{\mu^*}{\tan\beta} & M_{\tilde{U}}^2 + m_u^2 + D_{\tilde{u},R} \end{pmatrix}. \tag{1}$$

The most important terms with respect to our study are the soft mass matrices $M_{\tilde{Q}}^2$ and $M_{\tilde{U}}^2$, together with the trilinear coupling matrix $T_u$. The remaining parameters, not directly related to the squarks, are the Higgsino potential $\mu$, the up-type quark mass matrix $m_u$, and the ratio of the vacuum expectation values of the two Higgs doublets $\tan\beta = v_u/v_d$. Finally, the $D$-terms are given by $D_{\tilde{u},L} = m_Z^2 (T_u^3 - e_u s_W^2) \cos 2\beta$ and $D_{\tilde{u},R} = m_Z^2 e_u s_W^2 \cos 2\beta$, with $m_Z$ being the $Z$ boson mass, $s_W$ and $c_W$ are the sine and cosine of weak mixing angle and $T_u^3$ and $e_u$ the weak isospin and electric charge of the up-type quarks.

The underlying flavour structure enters the mass matrix through the soft mass and trilinear matrices. Under the assumption of Minimal Flavour Violation (MFV), these matrices are diagonal, such that the CKM-matrix remains the only source of quark flavour violation. Relaxing this assumption, i.e. considering the more general framework of Non-Minimal Flavour Violation (NMFV), allows for off-diagonal entries within these three matrices. Let us note that the same arguments and definitions hold in the sector of down-type squarks and sleptons, which are, however, beyond the scope of the present work.

From the up-type squark mass matrix in Eq. (1), the rotation to the basis of physical mass eigenstates $(\tilde{u}_1, \ldots, \tilde{u}_6)$ is done through

$$\text{diag}(m_{\tilde{u}_1}^2, \ldots, m_{\tilde{u}_6}^2) = \mathcal{R}^{\tilde{u}} \mathcal{M}_{\tilde{u}}^2 \mathcal{R}^{\tilde{u}\dagger}. \tag{2}$$

By convention, the mass eigenstates states $\tilde{u}_i$ ($i = 1, \ldots, 6$) are labelled to be crescent in mass. All information about the flavour structure of the up-type squarks is contained in the rotation matrix $\mathcal{R}^{\tilde{u}}$, and couplings involving up-type squarks in the physical basis relate to the entries of this matrix.

As discussed in the Introduction, a direct reconstruction of the complete squark rotation matrix cannot be aimed at in a near or even mid-term future. We therefore introduce a somewhat less precise but still very meaningful quantity, which is the stop flavour content of the

lightest up-type squark $\tilde{u}_1$. This quantity, defined as

$$x_{\tilde{t}} \equiv (\mathcal{R}^{\tilde{u}})^2_{13} + (\mathcal{R}^{\tilde{u}})^2_{16}, \tag{3}$$

will be at the centre of the present study. In order to sample the parameter space, we will in addition make use of the quantities $\theta_{\tilde{t}}$ and $\theta_{\tilde{c}}$, which correspond to the top and charm helicity mixing within the lightest state $\tilde{u}_1$, such that

$$
\begin{aligned}
\left(R^{\tilde{u}}\right)_{12} &= \sqrt{1 - x_{\tilde{t}}} \cos\theta_{\tilde{c}}, & \left(R^{\tilde{u}}\right)_{13} &= \sqrt{x_{\tilde{t}}} \cos\theta_{\tilde{t}}, \\
\left(R^{\tilde{u}}\right)_{15} &= \sqrt{1 - x_{\tilde{t}}} \sin\theta_{\tilde{c}}, & \left(R^{\tilde{u}}\right)_{16} &= \sqrt{x_{\tilde{t}}} \sin\theta_{\tilde{t}}.
\end{aligned}
\tag{4}
$$

The cases $x_{\tilde{t}} = 0$ and $x_{\tilde{t}} = 1$ correspond to MFV with respectively $\tilde{u}_1$ being a pure charm-flavoured or top-flavoured state. Moreover, $\cos\theta_{\tilde{t},\tilde{c}} = 0$ corresponds to a "right-handed" squark, while $\cos\theta_{\tilde{t},\tilde{c}} = 1$ corresponds to a "left-handed" squark.

## 3 Observables related to flavour violation at LHC

If a squark should be observed at the Large Hadron Collider or any future hadron collider, it will most likely be produced from (flavour-conserving) gluon-initiated processes and manifest through its decay into quarks and gauginos. In our setup, this corresponds to the decay modes

$$\tilde{u}_1 \to t\tilde{\chi}_1^0, \qquad \tilde{u}_1 \to c\tilde{\chi}_1^0, \qquad \tilde{u}_1 \to b\tilde{\chi}_1^+, \tag{5}$$

which are simoultaneously open if the squark is a mixture of the two flavours, i.e. if $0 < x_{\tilde{t}} < 1$. Here, the neutralinos manifest as missing transverse energy, while the charginos will decay further into $W$-bosons and neutralinos.

Our study is based on the assumption that these decays are observed, and that we have access to the observables

$$m_{\tilde{u}_1}, \quad m_{\tilde{\chi}_1^0}, \quad m_{\tilde{\chi}_1^+}, \quad R_{c/t} = \frac{\mathrm{BR}(\tilde{u}_1 \to c\tilde{\chi}_1^0)}{\mathrm{BR}(\tilde{u}_1 \to t\tilde{\chi}_1^0)}, \quad R_{b/t} = \frac{\mathrm{BR}(\tilde{u}_1 \to b\tilde{\chi}_1^+)}{\mathrm{BR}(\tilde{u}_1 \to t\tilde{\chi}_1^0)}. \tag{6}$$

Note that the production cross-section of the squarks, as well as their branching ratios alone, are difficult to access. We therefore choose to work with the ratios defined above rather than with the pure associated event rates. The mixed "top-charm" production channel at the LHC may be used to obtain the observable $R_{c/t}$, together with the standard "top-top" channel. Analytical expressions for the relevant decay rates in the NMFV framework can be found in Ref. [18]. Note that in the definition of the ratios $R_{c/t}$ and $R_{b/t}$, we assume without loss of generality that the decay into top quarks is always open.

For the further study, it is interesting to examine those expressions in order to find the $x_{\tilde{t}}$-dependence of the observables in certain limits concerning the nature of the involved neutralinos and charginos. For example, assuming a pure higgsino-like neutralino and neglecting the neutralino mass with respect to the squark mass, we obtain

$$R_{c/t}\Big|_{\tilde{\chi}_1^0 = \tilde{H}^0,\, m_{\tilde{u}_1} \gg m_{\tilde{\chi}_1^0}} = \frac{m_c^2}{m_t^2} \frac{1 - x_{\tilde{t}}}{x_{\tilde{t}}}, \tag{7}$$

As a second example, we assume a pure bino-like neutralino and obtain

$$R_{c/t}\Big|_{\tilde{\chi}_1^0 = \tilde{B}^0,\, m_{\tilde{u}_1} \gg m_{\tilde{\chi}_1^0}} = \frac{1 - x_{\tilde{t}} + \kappa_c \left(R^{\tilde{u}}\right)^2_{15}}{x_{\tilde{t}} + \kappa_t \left(R^{\tilde{u}}\right)^2_{16}} \longrightarrow \frac{1 - x_{\tilde{t}}}{x_{\tilde{t}}}, \tag{8}$$

Table 1: Scanned ranges of the parameters associated to the squark (left) and gaugino sector (right). All masses are given in GeV.

| Variable | Range |
|---|---|
| $m_{\tilde{u}_1}$ | $[700, 2000]$ |
| $x_{\tilde{t}}$ | $[0, 1]$ |
| $\cos \theta_{\tilde{t}}$ | $[0, 1]$ |
| $\cos \theta_{\tilde{c}}$ | $[0, 1]$ |

| Variable | Range |
|---|---|
| $M_1$ | $[600, 2000]$ |
| $M_2$ | $[600, 2000]$ |
| $\mu$ | $[600, 2000]$ |

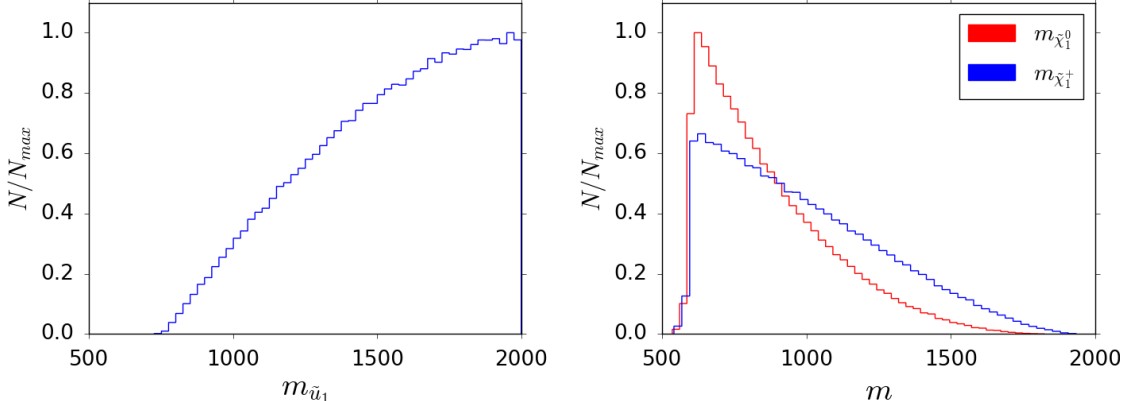

Figure 1: Distributions of the squark (left) and gaugino (right) masses obtained from the scan summarized in Table 1. The masses are given in GeV. The distributions show the number $N$ of points per bin normalized to the maximum value $N_{\max}$.

where $\kappa_q = e_q^2/\left(e_q - T_q^3\right)^2 - 1 = 15$ for $q = c, t$, and the last expression holds for a pure "left-handed" or a pure "right-haded" squark. Finally, for a pure wino-like neutralino, the ratio becomes

$$R_{c/t}\Big|_{\tilde{\chi}_1^0 = \tilde{W}^0} = \frac{B_c \, \lambda_c^{1/2} \left(R^{\tilde{u}}\right)_{12}^2}{B_t \, \lambda_t^{1/2} \left(R^{\tilde{u}}\right)_{13}^2} \quad \longrightarrow \quad \frac{B_c \, \lambda_c^{1/2}}{B_t \, \lambda_t^{1/2}} \frac{1 - x_{\tilde{t}}}{x_{\tilde{t}}}, \tag{9}$$

where $\lambda_q = m_{\tilde{u}_1}^4 + m_{\tilde{\chi}_1^0}^4 + m_q^4 - 2\left(m_{\tilde{u}_1}^2 m_{\tilde{\chi}_1^0}^2 + m_{\tilde{u}_1}^2 m_q^2 + m_{\tilde{\chi}_1^0}^2 m_q^2\right)$ denotes the usual Källén function associated to the squark decay and $B_q = m_{\tilde{u}_1}^2 - m_{\tilde{\chi}_1^0}^2 - m_q^2$ for $q = c, t$. Here, the last expression holds for a pure "left-handed" squark.

In order to gain a better understanding of these ratios, we start by randomly scanning over the parameters governing the lightest squark, neutralino, and chargino. More precisely, we vary the physical squark mass $m_{\tilde{u}_1}$, and the parameters $x_{\tilde{t}}$, $\theta_{\tilde{t}}$, and $\theta_{\tilde{c}}$ defining its flavour decomposition. In the gaugino sector, we vary the bino, wino, and Higgsino mass parameters $M_1$, $M_2$, and $\mu$. The physical gaugino masses are obtained by diagonalizing the mass matrices at the tree-level.

As the expressions in Eqs. (7) – (9) do not exhibit a dependence on $\tan \beta$, we conclude that this parameter only has a mild impact on the observables of our interest. We therefore fix $\tan \beta = 10$ throughout the presented analyses. All parameters are scanned over in a uniform manner according to the ranges given in Table 1. The corresponding parameter distributions are illustrated in Fig. 1 for the relevant physical masses and Fig. 2 for the corresponding mixing parameters, respectively. The shape of the mass distributions are explained by the fact that we require the decay modes mentioned above to be kinematically allowed, which favours larger squark and smaller gaugino masses. Since we impose a flat distribution of the stop content $x_{\tilde{t}}$, the elements $\left(\mathcal{R}^{\tilde{u}}\right)_{1i}$ ($i = 2, 3, 5, 6$) of the up-squark rotation matrix follow a parabolic

distribution. As the distributions of $\left(\mathcal{R}^{\tilde{u}}\right)_{1i}$ for $i = 3, 5, 6$ are similar to the one of $\left(\mathcal{R}^{\tilde{u}}\right)_{12}$, they are not shown separately in Fig. 2.

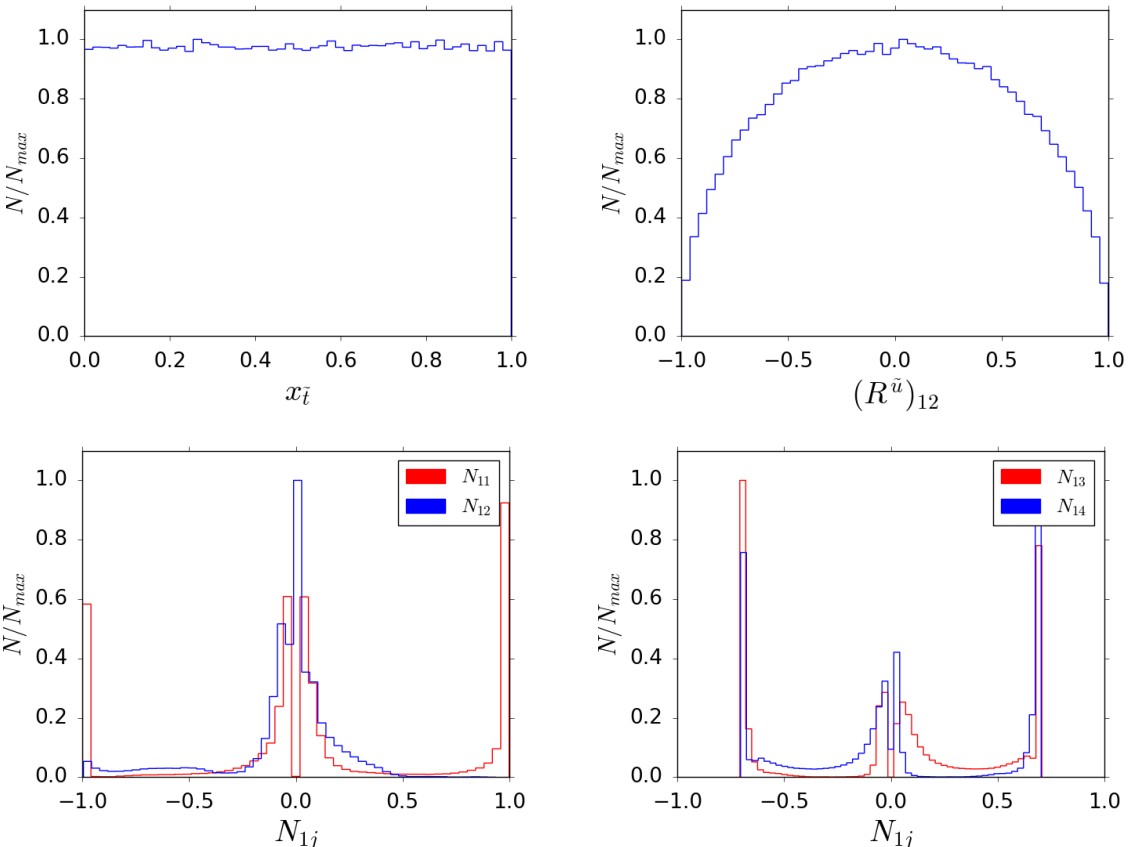

Figure 2: Distributions of the squark (upper row) and neutralino (lower row) mixing parameters associated to the masses shown in Fig. 1. The distributions are shown on a linear scale.

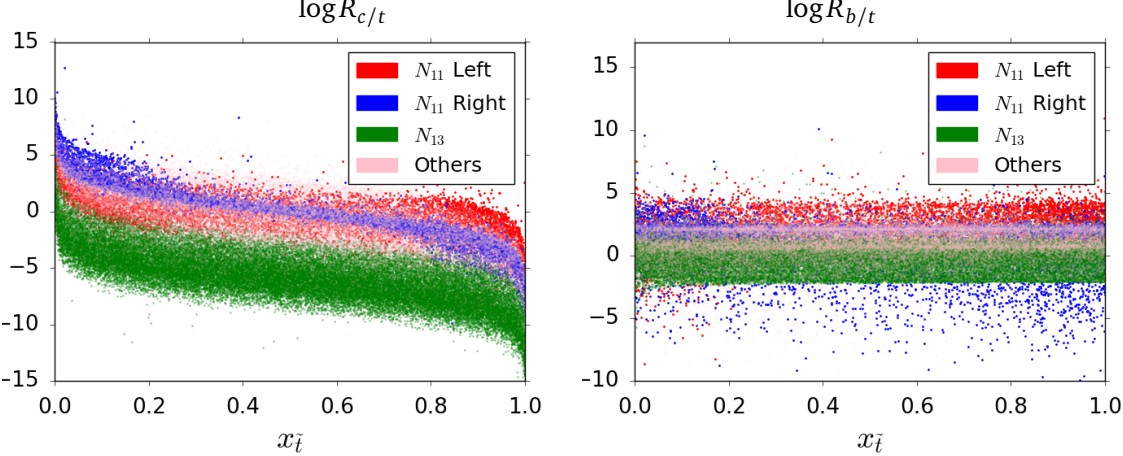

Figure 3: Distributions of the ratios $R_{c/t}$ (left) and $R_{b/t}$ (right) of the decay modes defined in Eq. (6) in dependence of the stop composition $x_{\tilde{t}}$ of the decaying squark. The colour code refers to different combinations of neutralino compositions and squark "chiralities".

For each parameter point, the gaugino masses and the ratios $R_{c/t}$ and $R_{b/t}$ of our interest are computed using the full analytical expressions of Ref. [18]. The results are depicted in Fig. 3, where we indicate as colour code the dominant component of the involved neutralino as well as the nature of the decaying squark. As expected from Eqs. (7) – (9), distinct regions are observed in the distributions of $R_{c/t}$. The same kind of feature appears for the ratio $R_{b/t}$. More precisely, the two ratios depend strongly on the neutralino decomposition and the "chirality" (expressed in terms of $\theta_{\tilde{t}}$ and $\theta_{\tilde{c}}$ defined in Eqs. (4)) of the decaying squark.

The width of each band in Fig. 3 is due to the fact that the majority of the parameter points feature mixed gauginos and squarks rather than corresponding to the limit cases discussed above. Nevertheless, the presence of the observed rather distinct regions is an important feature which will turn out to be crucial in the identification of the squark flavour decomposition from the observables given in Eq. (6).

## 4 Likelihood inference in a simplified model

In order to infer the stop component $x_{\tilde{t}}$ of the observed squark, we start by constructing a maximum likelihood estimator. For a given set of data,

$$D = \{m_{\tilde{u}_1}, m_{\tilde{\chi}_1^0}, m_{\tilde{\chi}_1^\pm}, R_{c/t}, R_{b/t}\}, \tag{10}$$

supposed to be obtained at the Large Hadron Collider, we associate a likelihood value to each point of an ensemble of random parameter points. Assuming uncorrelated parameters and thus a Gaussian distribution, this likelihood takes the form

$$\ln \mathcal{L}(\theta) = -\frac{1}{2} \sum_i \left( \frac{\theta_i - D_i}{\sigma_i} \right)^2, \tag{11}$$

with $\theta$ being the set of parameters associated to the parameter point under consideration and $\sigma_i$ being the error associated to the observable $D_i$. Even if in practice the parameters of interest are correlated, a Gaussian distribution constitues a reasonable approximation, as will be seen in the following.

We now divide the interval $x_{\tilde{t}} \in [0; 1]$ into $N$ bins of equal size. For each bin $j = 1, \ldots, N$, we then compute the average likelihood $\hat{\mathcal{L}}_j(x_{\tilde{t}})$ of all random parameter points having their value of $x_{\tilde{t}}$ inside the given bin. From the obtained values of $\hat{\mathcal{L}}_j(x_{\tilde{t}})$ over the interval $x_{\tilde{t}} \in [0; 1]$, we can fit a Gaussian distribution in order to find the maximum of likelihood corresponding to the inferred value of the stop component $x_{\tilde{t}}$. The associated uncertainty $\sigma(x_{\tilde{t}})$ is then based on the standard deviation value of the Gaussian fit.

As a first step, for the sake of simplicity, and in order to illustrate the proposed inference method, we fix the parameters associated to the neutralino and chargino decomposition as

$$N_{1l} = 0.5, \qquad U_{11} = V_{12} = 1, \qquad U_{12} = V_{11} = 0, \tag{12}$$

where $N$, $U$, and $V$ denote the mixing matrices associated to the neutralinos and charginos. In other words, we consider a maximally mixed neutralino. For the present example, we have performed a random scan over the five parameters of Eq. (10) leading to an ensemble of $5 \cdot 10^8$ parameter points. Moreover, we assign a common value of $\sigma_i = 0.25 D_i$ to the uncertainties entering the likelihood calculation.

Assuming four different test parameter points $P_i$ ($i = 1, \ldots, 4$) representing different configurations, we perform the analysis described above and infer the stop component $x_{\tilde{t}}$ using a Gaussian likelihood fit. The results are illustrated in Fig. 4 and summarized in the upper part of Table 2. More precisely, for each test parameter point, we show in Fig. 4 the average

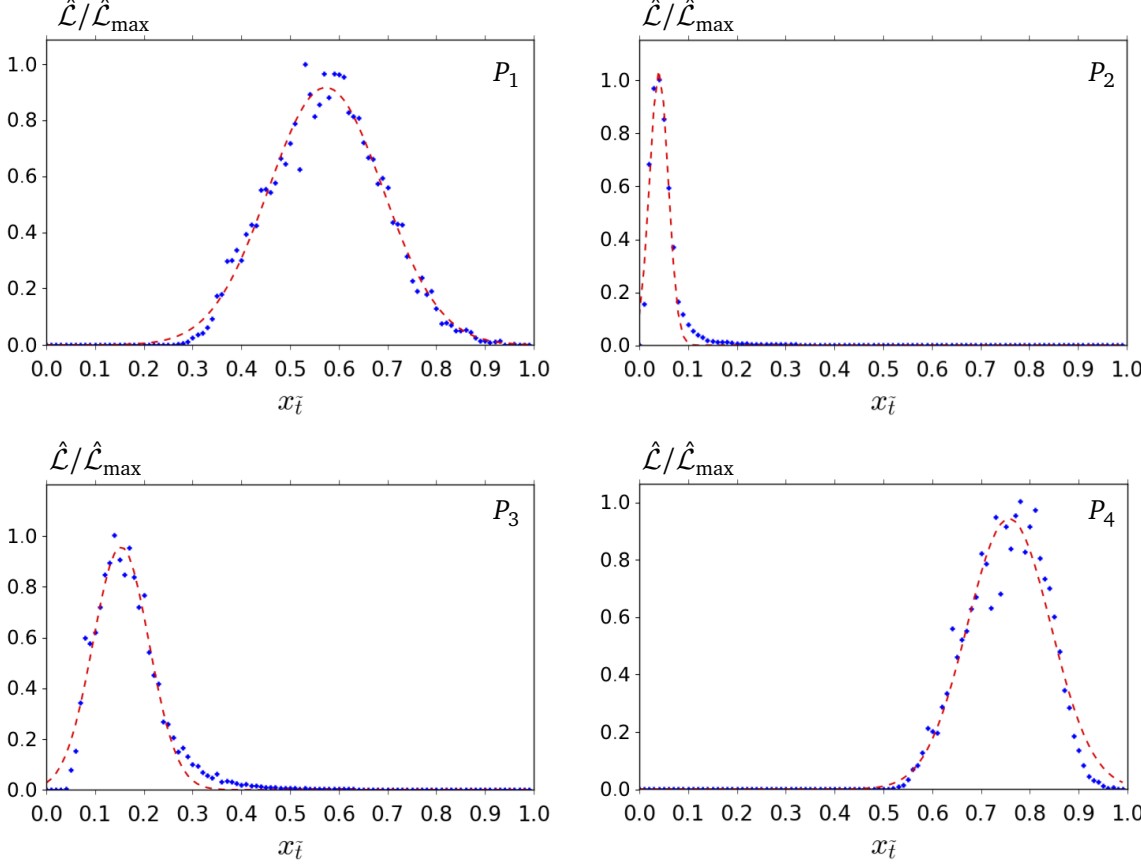

Figure 4: Likelihood fit for four test data sets featuring a fixed gauginos composition as in Eq. (12). The resulting inferred values of the stop component are listed in Table 2. The distributions show the averaged likelihood $\hat{\mathcal{L}}$ normalized to the maximum value $\hat{\mathcal{L}}_{\max}$.

likelihood $\hat{\mathcal{L}}_j(x_{\tilde{t}})$ obtained for each bin together with the Gaussian fit. As can be seen, our method manages to recover the actual stop component within the resulting uncertainty from the Gaussian fit.

As second and final step, we relax the assumption on the gaugino decompositions given in Eq. (12), and include the gaugino mixing parameters in the random scan. Again, we generate an ensemble of $5 \cdot 10^8$ parameter points with $\sigma_i = 0.35 D_i$, and apply our reconstruction method to two data sets $P_5$ and $P_6$. The results are shown in Fig. 5 and summarized in the lower part of Table 2. Even if the true stop components lie within the infered intervals, the uncertainties are much larger in this case, such that the results may become meaningless in certain cases. In addition, from Fig. 5 we can see that the likelihood is no longer Gaussian. This is due to the fact that here different regions of the parameters present a concentration of points able to explain the data.

Let us briefly discuss the impact of the uncertainties, which we have investigated by varying the value of $\sigma_i$ $(i = 1, \ldots, 5)$ for a given reference point. As it can be expected, increasing the uncertainties $\sigma_i$ leads to an increase in the uncertainty $\sigma(x_{\tilde{t}})$ obtained from the Gaussian fit. However, special care has to be taken when reducing the value of $\sigma_i$. First, the quality of the Monte Carlo sampling plays a crucial role. Indeed, if the parameter space is not populated well enough, the Gaussian fit "breaks down", i.e. cannot yield a meaningful result. Second, if one considers the more general setup, e.g., without fixing the gaugino parameters, degeneracies between the observables and the top-content $x_{\tilde{t}}$ appear, as can be seen in Fig. 3. This may lead to additional complications concerning the treatment of uncertainties.

Table 2: Parameters of the test data sets together with the assumed relative error $\sigma_i/D_i$ and the stop component obtained from the likelihood fits illustrated in Figs. 4 – 5. All masses are given in GeV.

| Data set | $m_{\tilde{u}_1}$ | $m_{\tilde{\chi}_1^\pm}$ | $m_{\tilde{\chi}_1^0}$ | $x_{\tilde{t}}$ | $\sigma_i/D_i$ | inferred $x_{\tilde{t}} \pm \sigma(x_{\tilde{t}})$ |
|---|---|---|---|---|---|---|
| $P_1$ | 1015.73 | 699.60 | 604.39 | 0.66 | 0.25 | $0.57 \pm 0.16$ |
| $P_2$ | 1798.29 | 303.02 | 267.66 | 0.04 | 0.25 | $0.04 \pm 0.03$ |
| $P_3$ | 1488.78 | 321.53 | 244.21 | 0.08 | 0.25 | $0.15 \pm 0.08$ |
| $P_4$ | 1422.50 | 1001.11 | 637.85 | 0.83 | 0.25 | $0.76 \pm 0.12$ |
| $P_5$ | 1369.07 | 281.13 | 276.32 | 0.04 | 0.35 | $0.03 \pm 0.03$ |
| $P_6$ | 1770.52 | 717.95 | 511.39 | 0.65 | 0.35 | $0.00 \pm 0.90$ |

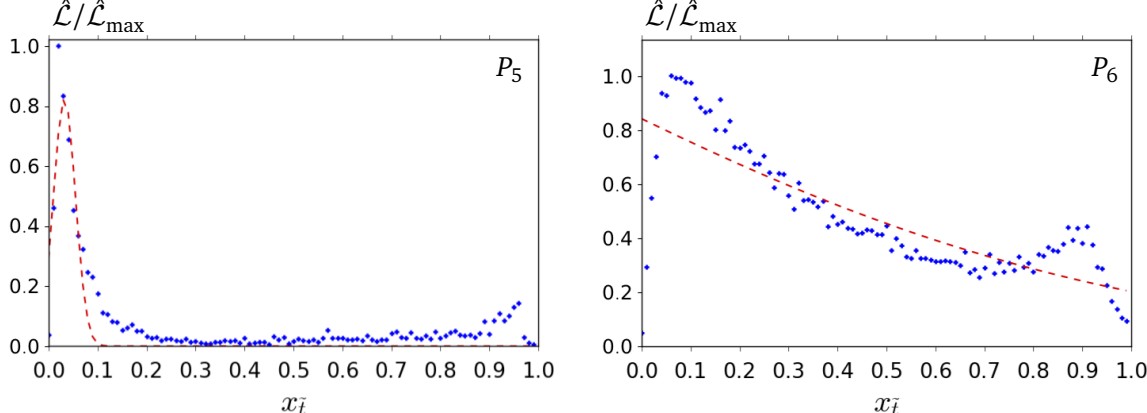

Figure 5: Same as Fig. 4 for two test parameter points obtained by scanning in addition over the parameters related to the gaugino sector.

In this first attempt of reconstructing the top-content $x_{\tilde{t}}$, we do not perform a dedicated analysis of the impact of the uncertainties $\sigma_i$. However, this question will need to be addressed properly in the case of an actual observation of a squark-like state. In this situation, the analysis proposed here will become crucial, and information about the underlying uncertainties will be known.

The uncertainties associated to the ratios $R_{c/t}$ and $R_{t/b}$ will be the most limiting factors of the analysis. In particular, $R_{c/t}$ is the most constraining observable, since it shows a strong correlation with the parameter $x_{\tilde{t}}$, as can be seen in Fig. 3. As a last comment, let us emphasize that the observables $D_i$ should have different relative uncertainties $\sigma_i$.

We conclude that the present method is not suitable if no additional independent knowledge on the gaugino sector, nor other relevant observables, are available. Here, we do not aim at studying the limit of the present method associated to the quality of the parameter space sampling, which will be necessary for a concrete analysis rather than for the simplified setup under consideration here.

## 5 Multivariate analysis in a simplified model

In order to go beyond the likelihood inference presented in the previous Section, especially in a more realistic setup such as, e.g., the more complete Minimal Supersymmetric Standard Model (MSSM) discussed in Ref. [35], we now employ a multivariate analysis (MVA) classifier. We start by presenting results obtained from a multi-layer perceptron (MLP) provided by ROOT

through the TMVA package [54] for the simplified setup already used in Secs. 3 and 4. The discussion of the complete MSSM with squark generation mixing of Ref. [35] will follow in Sec. 6.

In this context, the goal of our analysis is slightly different with respect to the previous Section. While the likelihood inference aims at estimating the actual stop component of the observed squark, a multivariate analysis is designed to effiently classify different configurations. In order to provide a simple illustration, we define two categories based on the stop composition $x_{\tilde{t}}$, which remains the key quantity of our interest. We will divide the parameter space into "top-flavoured" squarks and "charm-flavoured" squarks according to

$$x_{\tilde{t}} < 0.5 \quad \Longleftrightarrow \quad \text{"charm} - \text{flavoured"}, \tag{13}$$

$$x_{\tilde{t}} > 0.5 \quad \Longleftrightarrow \quad \text{"top} - \text{flavoured"}. \tag{14}$$

Let us note that these categories are for the moment rather arbitrary and aim at the illustration of the method rather than representing specific physical regions. In particular, additional categories can be defined in order to refine the analysis. Such a case will be discussed in Sec. 6. Based on the two categories, the MLP can be trained on the parameter points obtained from a random scan, and subsequently tested on a subset of points, the test sample, in order to compute the efficiency and the misidentification rate of the classifier. The analysis presented here is based on a training sample of $10^6$ points, which have been obtained by uniformly scanning as indicated in Table 1.

The classifier basically combines the set of obervables given in Eq. (6), i.e. $m_{\tilde{u}_1}$, $m_{\tilde{\chi}_1^0}$ $m_{\tilde{\chi}_1^+}$, $R_{c/t}$, and $R_{b/t}$ into a single variable, the so-called MLP response comprised between 0 and 1. The algorithm will associate an MLP value to each parameter point of the scan, depending on the set of observables that maximize the separation between the two categories. The obtained MLP responses will be presented as a histogram containing the distributions associated to the two categories to be seperated. If the MLP is rather efficient, the two distributions peak at the extremities 0 and 1, respectively.

A key point of such an analysis is the danger of so-called "overtraining", meaning that training the algorithm on a too small dataset may enforce the identification of unphysical regions, i.e. statistical fluctuations, as physical ones. We have performed an overtraining check by comparing the classification performance on the training sample and on the test sample. The behavior of the algorithm being the same on the two samples, we conclude that there are no statistical fluctuations having an impact on the classification.

The rather simple situation of having only two categories will also serve to study the influence of the underlying prior distribution, in particular of the stop component $x_{\tilde{t}}$. We start from the same setup as in Sec. 3, where the random parameter scan has been performed such that the stop component $x_{\tilde{t}}$ exhibits a flat distribution. For this case, we show the obtained MLP response for the two categories in Fig. 6, together with the prior distribution of the stop component (see also Fig. 2). If a set of observables leads, e.g., to an MLP response close to 1, the parameter point is likely to belong to the category of "charm-like" stages ($x_{\tilde{t}} < 0.5$, shown in red), while for MLP responses close to 0, the associated points are likely to belong to the "top-like" category ($x_{\tilde{t}} > 0.5$, shown in blue). The ratio "top-like" over "charm-like" is quite large for small MLP values, while the opposite ratio is large for high MLP responses. Note that the histograms are presented on a logarithmic scale.

In the present case, the classifier manages to seperate the two categories with a rather good efficiency. For a given misidentification rate, the associated efficiency, i.e. the number of points of a chosen class surviving the misidentification cut, of the classifier can be computed based on a cut on the MLP response. To give an example, the efficiency for the "charm-like" (red) category is obtained as the ratio of the "charm-like" area above the cut and the total "charm-like" area. The cut is chosen such that the ratio of the "top-like" (blue) area over the

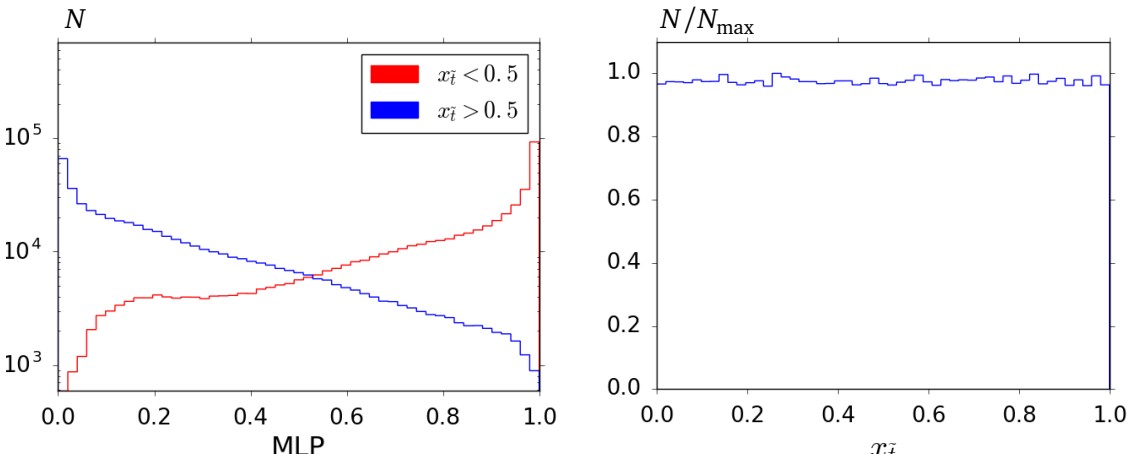

Figure 6: MLP response (number of points $N$, left panel) on the simplified scan based on a uniform prior (number of points $N$ normalized to the maximum value $N_{\text{max}}$, right panel) of the stop component $x_{\tilde{t}}$. The colour code corresponds to the seperation of "top-like" (blue) and "charm-like" (red) squarks.

"charm-like" (red) area above the cut corresponds to the misidentification rate imposed for the "charm-like" (red) category. It is to be noted that decreasing the misidentification rate (by increasing the cut value) will lead to a decrease of the efficiency. The efficiency for the "top-like" category is analogously obtained considering the corrresponding areas below a cut on the MLP response.

Here, for a misidentification rate of 10%, we obtain an efficiency of 54% for the "top-like" squark region and of 64% for the "charm-like" case. In other words, we can tag respectively approximately 54% and 64% of the points at 90% confidence level.

As a second example, we employ the classifier to the case of a non-uniform prior distribution of the stop-content $x_{\tilde{t}}$. Inspired by the results of Ref. [35], we choose a prior distribution peaking at its "MFV-like" extremities $x_{\tilde{t}} \approx 0$ and $x_{\tilde{t}} \approx 1$. Apart from the prior distribution (and thus the squark rotation matrix elements), the sample has the same characteristics as the previous one. The prior distribution and the resulting MLP response are shown in Fig. 7. While it is approximately symmetric in the case of a flat prior, the MLP response associated to the two categories is clearly non-symmetric in the present case. This can be traced to the fact that the observables used to classify are non-symmetric with respect to "top-flavoured" and "charm-flavoured" squarks.

In this example, for the misidentification rate of 10%, we obtain an efficiency of 64% for the "top-flavoured" category and an efficiency of 60% for the "charm-flavoured" category. It appears that the efficiency depends on the prior distribution. More precisely, considering the more peaked prior, the classifier becomes more efficient in identifying the "top-flavoured" category, but slighly less performant concerning the "charm-flavoured" category.

The increasing classification power coming from the prior distribution can intuitively be understood as the two categories are now more different. The border between the two cases, i.e. $x_{\tilde{t}} \sim 0.5$, where it is phenomenologically difficult to assign a given point to a single category, are less populated in the second case with non-uniform prior. It is therefore easier to maximize the separation. As a final comment, we would like to emphasize that the prior dependence is not a limitation of the present method, but a feature that the user should be aware of. After this first analysis within the simplified setup, we now aim at applying the MLP method to a more complete model.

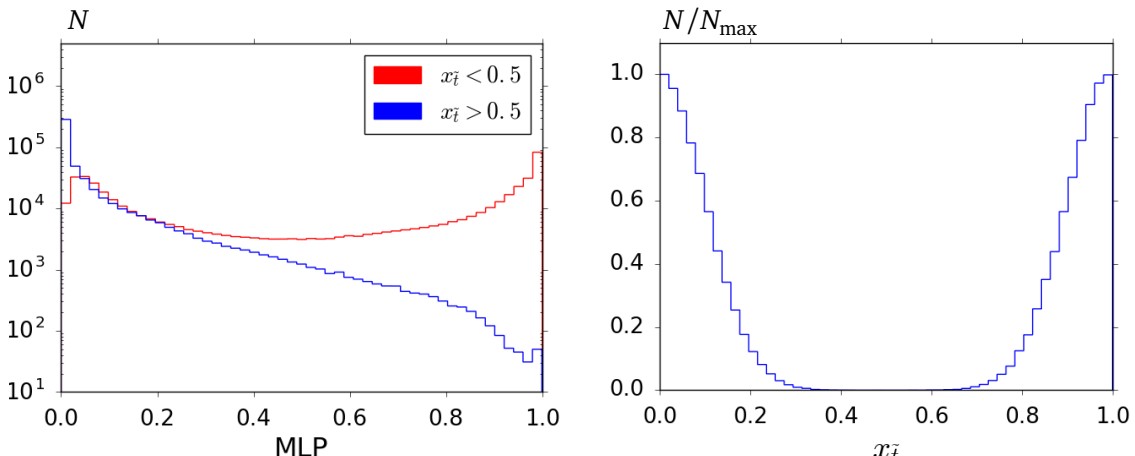

Figure 7: Same as Fig. 6 for an example of a non-uniform prior of the stop component $x_{\tilde{t}}$.

# 6 Application to the MSSM with mixed top-charm squarks

As announced in the previous Section, we finally apply the multivariate analysis (MVA) classifier to the Minimal Supersymmetric Standard Model (MSSM) with non-minimal flavour mixing between charm- and top-flavoured squarks. In order to work with a rather "realistic" setup, as basis of our study we choose to use the parameter points obtained in Ref. [35] by means of a Markov Chain Monte Carlo (MCMC) algorithm. These parameter points defined at the TeV scale have been shown to fulfill all relevant constraints coming from flavour and precision measurements, in particular the Higgs-boson mass, the decays $B \rightarrow X_s \gamma$ and $B \rightarrow X_s \mu\mu$, and the meson oscillation parameter $\Delta M_{B_s}$, to name the most relevant ones. For all details on the applied constraints and the related MCMC study of the MSSM with non-minimal flavour violation in the squark sector, the reader is referred to Ref. [35].

Following the preliminary study of the simplified setup in Sec. 5, it is interesting to examine the prior distribution of the quantity that we want to address, i.e. the stop component $x_{\tilde{t}}$ of the lightest up-type squark. As can be seen from its representation in Fig. 8, the distribution strongly peaks at the "MFV-like" ends. Moreover, flavour and precision data tend to favour a high charm content with respect to top content in the lightest squark. Note that this situation is similar to the non-uniform prior tested in Sec. 5, which turned out to yield a higher efficiency than the simpler uniform prior. However, in the present case, the prior distribution is non-symmetric between the MFV-like ends, the "charm-like" case being favoured.

Let us note that even in the case of such a peaked prior, the possibility of important flavour mixing is not ruled out. As a consequence, the question of identifying the flavour content of an observed squark is still of high interest. As discussed in Sec. 5, the prior distribution has an impact on the efficiency of the method, but not on its applicability. Finally, let us note that, although still relying on certain simplifications, the study of Ref. [35] is at our knowledge the most general phenomenogically analysis of the squark-flavour violating MSSM, and therefore the resulting parameter points represent a suitable sample to study in the given context.

We now perform the same MLP classification using a training sample containing about $6 \cdot 10^5$ points obtained from the MCMC analysis of Ref. [35] [1]. Starting from the prior distribution shown in Fig. 8, we divide the ensemble of points into four categories defined as

---

[1]For the present study, we have extended the sample resulting from the analysis presented in Ref. [35] using exactly the same computational setup.

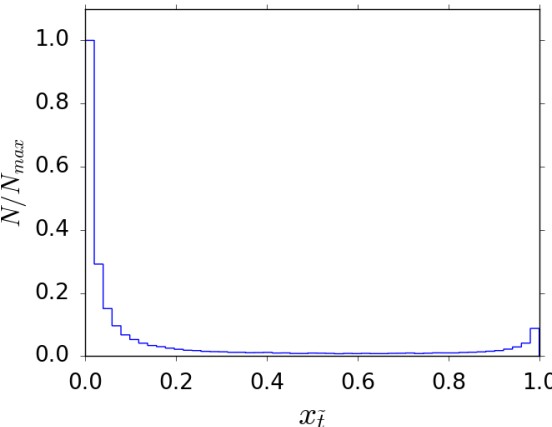

Figure 8: Prior distribution (nombre de points $N$ per bin normalized to the maximum value $N_{\mathrm{max}}$) of the stop composition $x_{\tilde{t}}$ from the MCMC analysis of Ref. [35].

follows:

$$
\begin{aligned}
0.00 \;\le\; x_{\tilde{t}} \;&<\; 0.05 \quad \Longleftrightarrow \quad \text{``charm MFV''} \\
0.05 \;<\; x_{\tilde{t}} \;&<\; 0.50 \quad \Longleftrightarrow \quad \text{``charm NMFV''} \\
0.50 \;<\; x_{\tilde{t}} \;&>\; 0.95 \quad \Longleftrightarrow \quad \text{``top NMFV''} \\
0.95 \;<\; x_{\tilde{t}} \;&\le\; 1.00 \quad \Longleftrightarrow \quad \text{``top MFV''}.
\end{aligned}
\tag{15}
$$

Note that, although the given definition of the above categories is again somewhat arbitrary, the exact value of the cuts between MFV and NMFV does not have a major impact on the methods presented in the following. It might, however, affect the efficiency of the proposed analysis, and the exact definition of the categories may in practice depend on the problem under consideration.

Here, we use the MVA classifier to seperate each of the four above categories from its complement, i.e. the ensemble comprising the three other classes. In Fig. 9, we show the MLP responses obtained for the four cases. As expected from the overpopulated prior region, the "charm MFV" category is rather well identified. However, the identification is less efficient for the two NMFV categories, which are underpopulated in the prior distribution. For the sake of a numerical comparison between the categories, and also to the cases presented in Sec. 5, we summarize the obtained efficiencies of the classifier in Table 3. In terms of physical interpretation, the efficiency of 95% for the "charm MFV" category is to be understood as follows: The probability to count an actual "charm MFV" parameter point correctly into this category is 95%, assuming that only 10% of the other parameter points (not belonging to this category) are wrongly classified as "charm MFV" (misidentification).

Overall, the performance of the classifier is better than for the simplified situations presented in Sec. 5. This can be traced to the underlying prior distribution of the stop content $x_{\tilde{t}}$ (see Fig. 8). The categories which are most difficult to identify, i.e. the two NMFV categories, are less populated in this particular model. The algorithm is therefore less performant in distinguishing these categories. The small bump observed around MLP $\sim 0.7 \ldots 0.8$ in both NMFV categories is an artefact of the employed multi-class MLP due to the presence of phenomenologically different regions.

Let us finally mention that we have also tested the likelihood inference method discussed in Sec. 4 on the present case of the NMFV-MSSM of Ref. [35]. However, for this method it turns out that inferring in a region of rather low density is quite difficult (contrary to the case of a uniform prior applied in Sec. 4). In addition, the strongly peaked prior distribution of the

Table 3: Efficiencies of the classification method for the four categories of our interest assuming a misidentification rate of 10%.

| Categories | | Efficiency |
|---|---|---|
| "charm" MFV | $0.00 \leq x_{\tilde{t}} < 0.05$ | 95% |
| "charm" NMFV | $0.05 < x_{\tilde{t}} < 0.50$ | 51% |
| "top" NMFV | $0.50 < x_{\tilde{t}} < 0.95$ | 41% |
| "top" MFV | $0.95 < x_{\tilde{t}} \leq 1.00$ | 69% |

stop component $x_{\tilde{t}}$ leads to a certain bias, such that the obtained results are not reliable any more. We therefore do not discuss this method further for the given model.

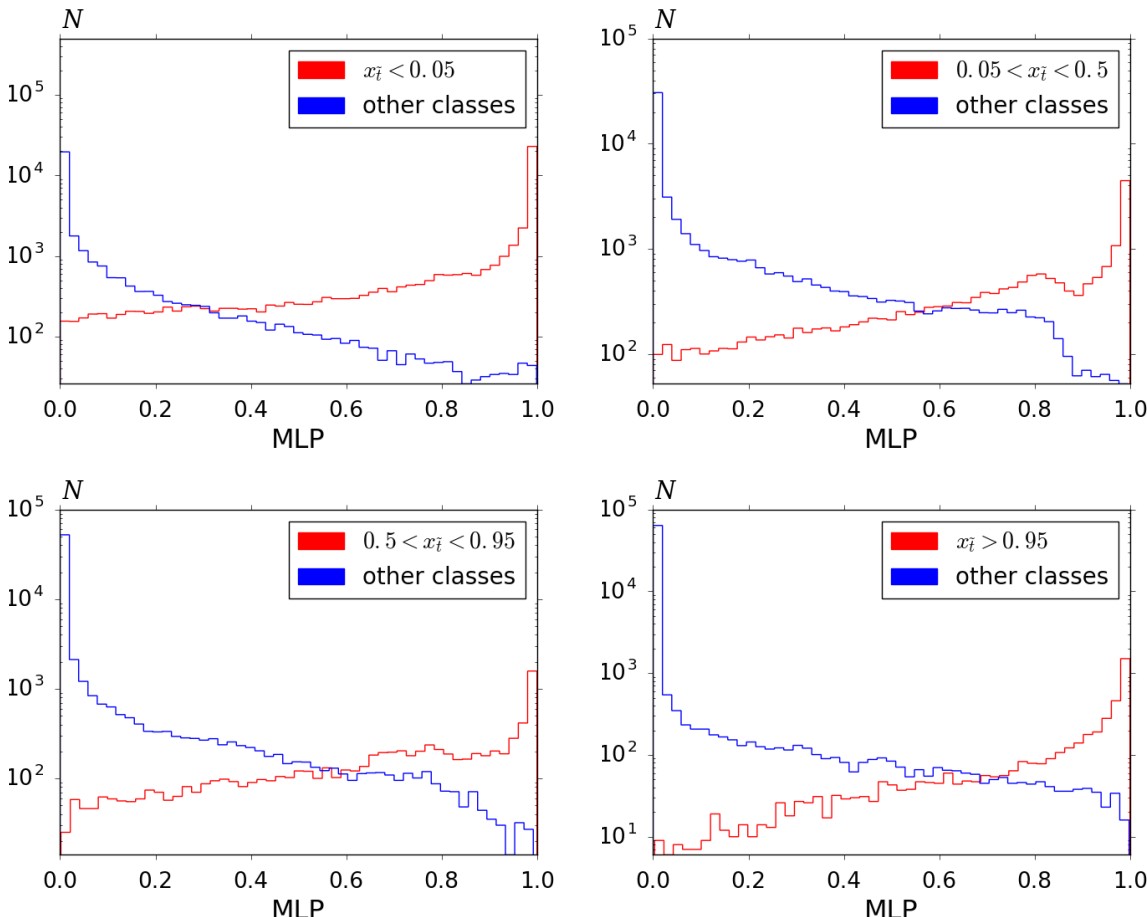

Figure 9: MLP response (number of points $N$) on the NMFV-MSSM of Ref. [35] for the seperation the "charm MFV" (upper left), "charm NMFV" (upper right), "top NMFV" (lower left), and "top MFV" (lower right) categories (red) from the remaining parameter points (blue).

## 7 Conclusion

We discuss the question to which extend the flavour decomposition of a squark-like state produced at the Large Hadron Collider can be reconstructed. As a starting point, we have considered a rather simple but typical set of collider observables related to inter-generational mixing

between top- and charm-flavoured squarks. The quantity of our interest is the top-flavour content of the observed squark state, since it may give valuable information on the flavour structure of the theory.

We first have employed a likelihood inference method, which basically allows to infer the top-flavour content of the observed squark. With the help of a simplified model incorporating non-minimal flavour violation between the top- and charm-flavoured squarks, we have obtained viable information on the squark flavour structure assuming that additional information, in particular concerning the gaugino sector, is provided. In absence of such information on the neutralino and chargino nature, the likelihood inference is less viable. However, the more additional information is available, e.g. on the gaugino sector (even if not fully determined), the more efficient this method will be. We also tried to use the likelihood inference method to the more general situation of the Minimal Supersymmetric Standard Model (MSSM) with additional top-charm mixing in the squark sector. However, it turns out to be inapplicable due to the somewhat extreme prior distribution of the top-flavour content and the available number of parameter points in the considered test sample based on previous work.

The second method consists of a multi-variate analysis classifier, which can efficiently separate two categories among a sample making use of a given set of observables. Performing this analysis on both the simplified setup and on the more general MSSM framework has led to promising results concerning the seperation between the Minimal and Non-Minimal Flavour Violation hypotheses. It turns out that this method can better deal with the strongly peaked prior distributions as it is the case in the considered MSSM with top-charm flavour mixing.

We want to emphasize the fact that the two methods are not addressing the same question. While the multi-variate analysis does not return an actual value for the top-flavour content of the squark, the likelihood inference can provide a reasonable estimation. However, the likelihood inference needs additional information, especially on the gaugino sector, and cannot handle very extreme prior distributions. These inconvenients can in turn be avoided by the use of the multivariate analysis, which already allows to gain valuable information on the flavour structure.

As this is a first attempt of the reconstruction of the squark flavour structure, the presented analysis relies on rather simple observables. Designing improved analyses inspired from this work should lead to a considerable improvement of the performances. As an example, one might consider additional observables related to the same parameters, such as, e.g., the top polarization from the squark decay or event rates stemming from gluino production and decay. From the machine-learning point of view, many algorithms exist for parameter-fitting problems and with a specific analysis it may be possible to access the actual value of the top-flavour content in a generic gaugino sector. Furthermore, considering new types of algorithms and additional observables may give access to the actual entries of the squark rotation matrix.

Since we did not assume any specific values for the masses nor any other observables in our scan of the parameter space, we show the feasibility of the proposed study in a generic way. For a concrete case, i.e. in case of an actual observation of a squark-like state at the LHC, this study has to be adapted to the actual signal. A more complete analysis of the proposed methods will therefore be in order. However, such an analysis, including in particular experimental details and uncertainties, is beyond the scope of the present Paper and will be necessary in order to render the proposed study well adapted to the actual observation. The experimental uncertainties fixed in our likelihood-based analysis of Sec. 4 can be adapted to the actual uncertainties associated to an observation. Concerning the multivariate analysis, the study proposed in Sec. 5 does not exploit the associated uncertainties. This will be rather technical to address and rely again on experimental knowledge associated to the actual observation.

## Acknowledgments

The authors would like to thank M. Reboud for useful discussions. The work of J.B. is supported by a Ph.D. grant of the French Ministry for Education and Research. This work is supported by *Investissements d'avenir*, Labex ENIGMASS, contrat ANR-11-LABX-0012. The figures presented in this Paper have been generated using MatPlotLib [55].

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
