# Peer review of "First steps towards the reconstruction of the squark flavour structure"

_SciPost Physics, doi:SciPost Phys. 6, 066 (2019)_

## Round 2 · Referee Report · Anonymous (Referee 1) · 2019-2-14

Strengths

see report

Weaknesses

see report

Report

The manuscript provides pioneering work for the determination of the flavour content of a squark from a set of experimental properties such as its mass and ratios of decay rates into final states with top, bottom or charm. Two complementary methods are employed, a likelihood inference method and a multivariate analysis, and their respective strengths and weaknesses are discussed. The work is original and the results are interesting and may trigger further studies. The presentation is clear and the style and length is appropriate.

I have one question to the authors to be addressed prior to publication: How would a change in the experimental uncertainties, assumed to be 25% for all observables entering the analysis, affect the results of the applied methods? Are the uncertainties in the ratios R_c/t and R_b/t the limiting factor in discriminiating between different flavour contents, as one might naively assume?

Requested changes

see report

  • validity: high
  • significance: high
  • originality: top
  • clarity: high
  • formatting: excellent
  • grammar: good

Author:  Björn Herrmann  on 2019-03-11  [id 462]

(in reply to Report 1 on 2019-02-14)
Category:
remark
answer to question

We would like to thank the Referee for the evaluation of our manuscript and for her/his interest in our work.

Regarding the questions concerning the uncertainties:

“I have one question to the authors to be addressed prior to publication: How would a change in the experimental uncertainties, assumed to be 25% for all observables entering the analysis, affect the results of the applied methods?”

During our analysis, we had investigated the question of the impact of the uncertainties mentioned by the Referee by varying the value of $\sigma_i$ for a given reference point.

As expected, increasing the uncertainties $\sigma_i$ leads to an increase in the uncertainty $\sigma(x_{\tilde{t}})$ obtained from the Gaussian fit.

However, special care has to be taken when reducing the value of $\sigma_i$. First, the quality of the Monte Carlo sampling plays a crucial role. Indeed, if the parameter space is not populated well enough, the Gaussian fit “breaks down”, i.e. it cannot yield a meaningful result. In addition, if one considers the more general setup, e.g. without fixing the gaugino parameters, degeneracies between the observables and the top-content $x_{\tilde{t}}$ appear, as can be seen in Fig. 3 of our manuscript. This may lead to additional complications concerning the uncertainties.

This is why, in this first attempt of reconstructing the top content $x_{\tilde{t}}$, we do not perform a dedicated analysis of the impact of the uncertainties $\sigma_i$. In the present work, we focus on showing that it is possible to obtain information on the flavour content (in particular $x_{\tilde{t}}$) rather than presenting a complete analysis including the uncertainties.

However, this question will need to be addressed properly in the case of an actual observation of a squark-like state. In this situation, the analysis proposed here will become crucial, and information about the underlying uncertainties will be known.

As a last comment, let us emphasize that the observables should have different uncertainties $\sigma_i$.

“Are the uncertainties in the ratios $R_{c/t}$ and $R_{b/t}$ the limiting factor in discriminiating between different flavour contents, as one might naively assume?”

The Referee is right by assuming that the uncertainties on $R_{c/t}$ and $R_{b/t}$ would be the most limiting factors in the analysis. In particular, $R_{c/t}$ is the most constraining observable, since it shows a strong correlation with the parameter $x_{\tilde{t}}$, as can be seen in Fig. 3 of our manuscript.

We will be happy to update our manuscript with the above comments, should it be judged necessary by the Referee or the Editor.

---

## Round 2 · Referee Report · Anonymous (Referee 2) · 2019-2-20

Strengths

1 - The study of the squark flavour structure can be potentially interesting in the case of discovery.

Weaknesses

1 - The analysis is very limited. It assumes the observation of model parameters at the Large Hadron Collider, not showing to which extent they are indeed accessible.
2 - The results need to be better presented. Several figures are challenging to interpret.

Report

The present manuscript addresses the determination of the squark flavour structure at the Large Hadron Collider (LHC). The proposed study explores two methods: a likelihood inference method and a multivariate analysis method to determine the top flavour content of a hypothetically observed squark particle.

The analysis infers the stop component, using the analytical expressions present in the literature, assuming that we will have experimental access to their leading dependencies $m_{\tilde{u}1}, \quad m{1}^0}, \quad m{1}^+}, \quad R} = \frac{{\rm BR}(\tilde{u1 \rightarrow c \chi}^0)}{{\rm BR}(\tilde{u1 \rightarrow t \chi, \quad R_{b/t} = \frac{{\rm BR}(\tilde{u}}^0)1 \rightarrow b \chi}^+)}{{\rm BR}(\tilde{u1 \rightarrow t \chi$}^0). The scope of the present study is very limited. In particular, it lacks a prediction for the experimentally accessible parameter space at the LHC for these observables, proving to which extent the proposed parameter space analysis is reliable. For example, the study strongly relies on the parameter $R_{c/t}$. This parameter can be challenging to obtain, making this study ineffective.

Furthermore, several results are not properly presented, rendering their interpretation challenging. For instance, several figures are plotted without tick marks and values in the y-axis to define a scale, units and y-axis label. See for instance Figs. 1, 2, 4, 5, 6, 7, and 8. This presentation does not meet the standards of a scientific publication.

Therefore, I cannot recommend this paper for publication.

Requested changes

1 - The scope of the current study is too limited. In particular, a proper LHC study, deriving the experimental sensitivity of the explored observables would be required to make this study robust.
2 - Figs. 1, 2, 4, 5, 6, 7, and 8 need a major revision, adding axis tick marks and values, units, and labels.

  • validity: low
  • significance: ok
  • originality: low
  • clarity: ok
  • formatting: acceptable
  • grammar: good

Author:  Björn Herrmann  on 2019-04-08  [id 490]

(in reply to Report 2 on 2019-02-20)

We would like to thank the Referee for reviewing our manuscript and providing us with the useful comments, which we hereby address:

1) We agree with the Referee that a more complete analysis will be in order, including experimental details and uncertainties. However, in our opinion, this is to be done in the case of an actual observation at the LHC or future collider, which would make the whole study necessary and well adapted to the observation. We did not assume any specific mass values nor other observables in our scan of the parameter space. We thus show the feasibility of the proposed study in a generic way. For a concrete case, i.e. in case of an actual observation, this study has to be adapted on the actual observation and its associated uncertainties. 

Concerning the Referee’s comment on the observable $R_{c/t}$, we agree that it may be experimentally challenging. However, as discussed in Ref. [37], the mixed “top-charm” final state can be reached via a dedicated search strategy. We base our study on the assumption that this strategy is employed, such that the corresponding part of parameter space, featuring sizeable generation mixing, is covered. This mixed “top-charm” channel may be used to obtain the observable $R_{c/t}$, together with the standard “top-top” channel.

2) We will be happy to improve the presented Figures, in particular providing axis ticks and labels (if missing) together with a measure of the shown probability densities.

We will be happy to modify our manuscript according to the above points, and provide a revised version.

---

## Round 2 · Referee Report · Anonymous (Referee 3) · 2019-3-25

Strengths

1 - The problem posed in the paper is interesting.
2 - The numerical analysis presented in the paper appears sufficient for an exploratory study.
3 - The paper itself is well structured and informative.

Weaknesses

1 - The motivation for doing the work described in the paper is not very much discussed.

2 - The description of the studies based on multivariate analysis is not very detailed, obscuring clarity of the presentation in that part of the paper.

Report

The goal of the paper is to study the potential of various reconstruction methods of a flavour composition of a squark, assuming its observation at the LHC. Two different methods are considered, the likelihood method and the multivariate analysis.

The numerical analysis presented in the paper appears to be done thoroughly. The obtained results would be important for determination of the flavour content of a squark, in the event of its discovery at the LHC. This is a first attempt of this kind. I would therefore consider the paper for publication, assuming the points below are addressed.

Requested changes

1 - The described work entirely relies on the assumption of existence of LHC measurements for the chosen set of five observables. Given null results from LHC searches to date, it seems a bold assumption to make. Therefore I wish the authors could justify it more and provide arguments why they believe the chosen observables can be still measured at the LHC at the accuracy level which would allow to perform studies described in the paper.

2 - The MLP analysis makes up a very substantial part of the work. However, its description leaves some open questions. In particular, it is not clear to me what the relation between the MLP response and the discrimination between the categories as well as the efficiency of the classifier is. For example, why the MLP response for the top-like (charm-like) category peaks on the left (right) in Fig. 6? And how are the MLP distributions shown in the plots related to the efficiency of the classifier, as implied by the sentence in section 5 starting with "As can be seen..."?

3 - Finally, one comment on notation: the matrices N,U and V are only referred to as mixing parameters without any further definition. Although for most readers it is clear what they are, a reference where their exact form can be found should be nevertheless given.

  • validity: good
  • significance: good
  • originality: high
  • clarity: good
  • formatting: perfect
  • grammar: perfect

Author:  Björn Herrmann  on 2019-04-08  [id 489]

(in reply to Report 3 on 2019-03-25)
Category:
remark
answer to question

We would like to thank the Referee for reviewing our manuscript and providing us with the useful comments, which we hereby address:

1) In the introduction of our manuscript, we state that recently it has been shown that current experimental limits (leading to the null result mentioned by the Referee) do not apply as they are to the case of non-negligible flavour mixing, see our Ref. [37]. In the maximal mixing case, squarks would even escape detection in current analyses. However, in Ref. [37], a complementary analysis is proposed which will cover the case of generation mixing in the squark sector. This strategy is based on the search for the mixed final state containing a top quark and a charm-flavoured jet plus missing energy stemming from the two associated neutralinos appearing in the squark decays. We therefore assume that such channels can be accessed at the LHC with sufficient luminosity, as discussed in Ref. [37]. 
The mixed “charm-top” channel may be used to obtain the observable $R_{c/t}$, together with the standard “top-top” channel. Note that $R_{c/t}$ is the observable having the largest impact on our study. However, we admit that we did not recast the analysis in order to check the expected accuracy of $R_{c/t}$. Such an analysis, involving mainly experimental knowledge, is beyond the scope of the present manuscript. 

The experimental uncertainties fixed in our likelihood-based analysis can be adapted to the actual observation of a squark-like state, and the associated decays. Concerning the multivariate analysis, the analysis proposed in our manuscript does not exploit the associated uncertainties. This will be rather technical to address and rely again on experimental knowledge associated to the actual observation, such that this point is beyond the scope of our paper.

2) The results of the MLP analysis are presented as histograms, in which a MLP value is computed for each point of the sample. If the MLP is rather efficient, the two distributions associated to the categories to be separated peak at the extremities (0 and 1). 

As an example, looking at Fig. 6, if a set of observables is leading to an MLP response of 1, the parameter point is very likely to belong to the “red” category, i.e. be a charm-like state, the ratio “red”/”blue” being very large. 

We can determine the efficiency of the classifier with respect to a certain misidentification rate. This is done by fixing a cut on the MLP value, defined such that the ratio of the “blue” area over the “red” area above the cut corresponds to the misidentification rate associated to the “red” category. The efficiency for the “red” category is then obtained as the ratio of the “red” area above the cut and the total “red” area. The same can of course be carried out for the “blue” category.

To give an example, in terms of physical interpretation, the “charm MFV” category given in Table 3: the “probability” to count an actual “charm MFV” parameter point into this category is 95%, assuming that 10% of the other parameter points (not belonging to this category) are classified as “charm MFV” (misidentification). 

Consequently, decreasing the misidentification rate (by changing the cut) will lead to a decrease of the efficiency.

3) The Referee is right that the matrices N, U, and V are not properly defined in our manuscript. We will add their definition to the manuscript.

We will be happy to modify and resubmit our manuscript according to the above points, in particular the motivation of our study and a clearer explanation of the MLP results.

---

## Round 3 · Author Response

Dear Sir or Madam,

we would like to thank the Referees for their careful evaluation of our manuscript as well as for their useful comments and questions.
We hereby submit a revised version of our manuscript, addressing the mentioned points.

Sincerely yours,
the authors

---

## Round 3 · List of Changes

We have made the following changes to our manuscript following the Editor's requests:

1) We have added a paragraph discussing the impact of the uncertainties on the likelihood-based analysis in Sec. 4. Moreover, the comments added in the Conclusion section refer to the treatment of uncertainties in the context of the multivariate analysis.

2) In the same paragraph within Section 4, we also comment on the uncertainties associated to the ratios R_{c/t} and R_{b/t}.

3) We have added a new paragraph in the end of the Conclusion section, explaining the lack of a more complete LHC analysis in the present paper. There is also a comment in the text added in Sec. 4.

4) The figures have been improved as requested. We have adapted the corresponding figure captions accordingly.

5) We have improved the discussion of the motivation in the Introduction section.

6) We have incorporated a clear explanation of the MLP results and their interpretation into our text. In particular, we have added a detailed explanation on pages 10 and 11. Moreover, on pages 14/15, we have added one sentence concerning the interpretation of the results presented in Table 3.

In addition to the recommendations formulated by the Referees and the Editor, we have incorporated the following minor modifications:

*) As one of the referees has pointed out, the matrices used to indicate the gaugino composition in Eq. (4.3) were not properly defined. We have included their meaning in the sentence following Eq. (4.3).

*) In Section 3, we have added a comment about how the key observable R_{c/t} may be obtained experimentally.

*) We have corrected a few minor typographical and punctuation errors.

---

## Editorial Decision

published